# Measuring facial mimicry: Affdex vs. EMG

**Jan-Frederik Westermann** *, **Ralf Schäfer, Marc Nordmann, Peter Richter, Tobias Müller, Matthias Franz**

Medical Faculty, Clinical Institute for Psychosomatic Medicine and Psychotherapy, University Hospital of the Heinrich-Heine-University, Düsseldorf, Germany

* westermannjan@yahoo.de

**Data Availability Statement:** All Data files are available from the Figshare database: Links: S1_Measuring Facial Mimicry Affdex vs. EMG_Psychometric Data https://figshare.com/articles/dataset/Untitled_ItemMeasuring_Facial_

## Abstract

Facial mimicry is the automatic imitation of the facial affect expressions of others. It serves as an important component of interpersonal communication and affective co-experience. Facial mimicry has so far been measured by Electromyography (EMG), which requires a complex measuring apparatus. Recently, software for measuring facial expressions have become available, but it is still unclear how well it is suited for measuring facial mimicry. This study investigates the comparability of the automated facial coding software Affdex with EMG for measuring facial mimicry. For this purpose, facial mimicry was induced in 33 subjects by presenting naturalistic affect-expressive video sequences (anger, joy). The response of the subjects is measured simultaneously by facial EMG (corrugator supercilii muscle, zygomaticus major muscle) and by Affdex (action units lip corner puller and brow lowerer and affects joy and anger). Subsequently, the correlations between the measurement results of EMG and Affdex were calculated. After the presentation of the joy stimulus, there was an increase in zygomaticus muscle activity (EMG) about 400 ms after stimulus onset and an increase in joy and lip corner puller activity (Affdex) about 1200 ms after stimulus onset. The joy and the lip corner puller activity detected by Affdex correlate significantly with the EMG activity. After presentation of the anger stimulus, corrugator muscle activity (EMG) also increased approximately 400 ms after stimulus onset, whereas anger and brow lowerer activity (Affdex) showed no response. During the entire measurement interval, anger activity and brow lowerer activity (Affdex) did not correlate with corrugator muscle activity (EMG). Using Affdex, the facial mimicry response to a joy stimulus can be measured, but it is detected approximately 800 ms later compared to the EMG. Thus, electromyography remains the tool of choice for studying subtle mimic processes like facial mimicry.

## 1 Introduction

Mimicry describes the imitation of facial expressions, intonation, and body posture between two interaction partners [1]. Facial mimicry is of particular importance because it has a specific emotional meaning as it represents a congruent mimic response to an emotional facial expression [2]. It is detectable in the electromyogram (EMG) after only 200–400 ms [3, 4] and occurs unconsciously and automatically [5]. Facial mimicry has been shown to be triggered even when affective cues are perceived only unconsciously [4, 6].

Mimicry_Automated_Facial_Coding_vs_EMG_ Psychometric_Data/21777281 S2_Measuring Facial Mimicry Affdex vs. EMG_EMG Data https:// figshare.com/articles/dataset/Measuring_Facial_ Mimicry_Automated_Facial_Coding_vs_EMG_ EMG_Data/21777275 S3_Measuring Facial Mimicry Affdex vs. EMG_Affdex Data https:// figshare.com/articles/dataset/Measuring_Facial_ Mimicry_Automated_Facial_Coding_vs_EMG_ Affdex_Data/21777200 S4_Measuring Facial Mimicry Affdex vs. EMG_Code for Statistics in R https://figshare.com/articles/software/S4_ Measuring_Facial_Mimicry_Affdex_vs_EMG_ Code_for_Statistics_in_R/22639150.

**Funding:** The authors received no specific funding for this work.

**Competing interests:** The authors have declared that no competing interests exist.

The role of facial mimicry in the recognition of others' emotions is controversial. In a widely accepted concept, facial mimicry leads to emotional contagion through a feedback mechanism [7]. This is thought to improve affect perception and thus the ability to empathize. This concept has been further developed as part of an embodiment approach to emotion recognition [8]. According to this, facial mimicry facilitates the decoding of observed emotions [9]. This hypothesis is supported by the fact that it has been shown that emotion recognition can be impaired when the subject's facial mimicry is impaired. [10, 11]. Similarly, an increase in mimic muscle activity (e.g., due to a task such as biting on a pen or holding a pen with the lips), may result in lower accuracy in facial expression recognition. In contrast, there is also evidence that facial mimicry does not improve emotion recognition [12, 13] and no consistent evidence for the feedback hypothesis could be found in reviews [14]. A broader consensus exists for the assumption that mimicry has a positive influence on social relationships [15]. It has been demonstrated that facial mimicry occurs to varying degrees depending on the situation in which the subject finds himself. Thus, the probability for facial mimicry to occur is higher when there is a desire to cooperate and lower in a competitive situation [16, 17]. According to this assumption, facial mimicry can be understood as an affiliative behavior and supports the establishment and maintenance of interpersonal relationships.[18–20]. Interaction partners are perceived as more likable when they subtly imitate the others behavior [21]. This makes it easier for an individual to receive acceptance in a group. The individual thus satisfies his or her need to belong to the group and, at the same time, the collective achievement of relevant goals is facilitated [22]. Facial mimicry occurs with varying likelihood depending on the valence of the affect. Consistent with an attachment-reinforcing function, smiling is more frequently imitated as an expression of a happy facial expression [20]. In encounters between strangers, a frown (7%) is imitated significantly less often than a smile (53%) [23].

The gold standard for measuring facial mimicry is EMG measurement of affect-relevant mimic target muscles. Measurement of EMG activities of the zygomaticus and corrugator muscles is commonly used to distinguish between hedonic and anhedonic affects [24–26]. Activation of the corrugator muscle results in a frown and the contraction of the zygomaticus muscle results in a smile. Dimberg [24] showed that the presentation of happy faces led to an increase in EMG activity in the zygomaticus muscle and the presentation of angry faces led to an increase in EMG activity in the corrugator muscle. While zygomaticus and corrugator muscles are good indicators of the valence of mimicry, it has been shown that many emotions are subject to a specific pattern of muscle activity and that this pattern is also reflected in the mimicry response [27].

It has also been shown that the mimic response to angry and happy faces results in visible congruent changes in facial expression [28]. This allows to measure facial mimicry using the Facial Action Coding System (FACS) [29]. The FACS is currently the most comprehensive method for coding facial expressions. Using videotaped faces, specially trained human coders can encode so-called action units. According to FACS, there are a total of 44 action units, with each action unit describing a specific facial mimic activity. Ekman assumes that certain combinations of action units can be used to infer the basic affects of fear, disgust, joy, sadness, surprise, and anger. From this, Ekman derived his own coding system, the Emotional Facial Action Coding System (EMFACS) [30]. Although EMFACS has never been published in a peer review process, it is widely used.

Besides the aspects mentioned above, the measurement of facial mimicry is also relevant from a clinical perspective. Some mental disorders that lead to impaired interpersonal communication and thus to distress and further mental comorbidities are associated with altered facial mimicry. For example, slowed [31] or decreased [32] facial mimicry has been demonstrated in patients with autism on EMG. Individuals with alexithymia [33] and Parkinson´s

disease [34] also exhibit a reduced facial mimicry response. Other mental disorders with impaired facial mimicry include schizophrenia [35, 36], borderline personality disorder [37], and depression [38]. The extent to which impaired facial mimicry moderates the severity and distress of these disorders is debated.

As already described, the measurement of facial mimicry is technically demanding. Measurement by EMG requires a complex measuring apparatus and experience in the application and interpretation of EMG signals. The analysis of video footage by FACS requires specially trained FACS raters and is very time-consuming. Recent methods for machine-learning assisted videographic measurement of mimic activity promise time-efficient and easy-to-interpret analysis. This could open up a large area of application in affect research.

The software Affdex (developed by Affectiva) investigated in this study is used on the iMotions platform. Affdex is one of the most widely used automated facial coding software. It promises ease of use and synchronization with other psychophysiological measures. It can be used to synchronously measure and evaluate various psychophysiological signals. Should a validation for the measurement of facial mimicry be successful, complex experimental paradigms could thus be performed in a relatively user-friendly manner. Affdex is based on a machine learning principle. A database of approximately 27,000 human FACS-encoded videos of affect-expressive faces is available. To evaluate the likelihood of activity of action units of new videos, Affdex compares them with the database [39]. In a further step, the combined activity of specific action units is used to derive the probability of the presence of a basic affect based on EMFACS [39]. However, the exact operation of the underlying algorithm is not disclosed, making it difficult for researchers to examine the software in detail.

There are already some studies available that have investigated different Automated Facial Coding (AFC) software concerning to certain features. Stöckli et al. [40] compare Affdex with the AFC software Facet and conclude that AFC has difficulties in detecting subtle affects. In another study, subjects were asked to mime happy and angry faces while both Affdex and EMG activity of zygomaticus and corrugator muscle were measured. A high positive correlation was found between the probability of joy and zygomaticus muscle activity and between anger and corrugator muscle activity [41]. While strong prototypical affect expressions were measured here, Höfling et al. [42] compare the ability of the AFC software Facereader (Noldus) with EMG to measure subtle affect expressions. Here, subjects were not asked to imitate the affect stimuli, but to behave passively. There was a congruent EMG activity for anger and joy, indicating a facial mimicry response, whereas the Facereader software had difficulty measuring the negative valence for the anger stimuli.

The ability of Affdex to measure facial mimicry has not yet been investigated. Furthermore, there have been no studies to date on the extent to which the AFC measurement for the lip corner puller and brow lowerer action units correlate with the EMG activity of the zygomaticus and corrugator muscles. This question is of interest because these muscles represent the underlying anatomical structures for the action units. The present study attempts to answer these open questions.

The aim of this study is to compare electromyography with the Affdex AFC software for measuring facial mimicry response to angry and happy faces.

For this purpose, a healthy cohort was shown videos of faces dynamically accumulating affect over time. The stimulus material consisted of video sequences of adult faces showing the basic affects of anger and joy. EMG measurements of the zygomaticus and the corrugator muscles and Affdex measurements were performed simultaneously. Subsequently, EMG activity of the zygomaticus and corrugator muscles was directly compared to the FACS-oriented Affdex action units lip corner puller and brow lowerer [43]. This allows for a direct comparison of measurement sensitivity, as these action units represent the visible correlates to the underlying

muscles [29]. Affdex uses additional information from the face for the measurements besides the lip corner puller and brow lowerer action units. Therefore, EMG activity was additionally compared with affect probabilities for joy and anger measured by Affdex.

We expected a positive correlation between the EMG-activity of zygomaticus muscle and action unit lip corner puller for affect joy. We also expected a positive correlation between the EMG-activity of corrugator muscle and action unit brow lowerer for affect anger [41]. However, there is also evidence that measuring subtle affect expressions may be more difficult for the Affdex software [40, 42].

## 2 Materials and methods

This study is part of a study project on differences in facial responsiveness of alexithymic and non-alexithymic subjects using EMG and Affdex [33]. In the study project, the mimic responses to video sequences with dynamically animated affective facial expressions of adults and children (anger, joy, disgust, surprise, sadness) were investigated. In the present study, only the response of the healthy control group to the adult stimuli of joy and anger was examined. The extent to which the EMG activity of the zygomaticus and corrugator muscles correlated with the Affdex measurements of lip corner puller, joy, brow lowerer, and anger was investigated.

### 2.1 Psychometric instruments

To ensure a psychologically healthy subject sample, exclusion criteria were screened by two structured interviews (Structured Clinical Interview for DSM-IV (SCID), Toronto Structured Interview for Alexithymia (TSIA)), questionnaires (Short version of the Autism Spectrum Quotient (AQ-short), Beck Depression Inventory II (BDI-II), Patient Health Questionnaire (PHQ-9), 20-item Prosopagnosia Index (PI-20), Toronto Alexithymia Scale (TAS-20)), and functional tests in the laboratory immediately before the start of the experimental part of the study.

The Structured Clinical Interview for DSM-IV (SCID) [44] used to identify psychiatric diagnoses according to the Diagnostic and Statistical Manual Fourth Edition (DSM-IV). The SCID is divided into two parts. The first part captures DSM-IV Axis I disorders (SCID-I) and the second part captures DSM-IV Axis II personality disorders (SCID II). In this study, only schizoid personality traits were recorded for the SCID-II, thus excluding schizoid traits in the subjects.

The Toronto Structured Interview for Alexithymia (TSIA) [45] is an instrument used for clinical and scientific purposes to identify alexithymic disorders, the regulation and processing of affects. In each case, the respondent is asked to name a corresponding situation from his or her life for different cases. A detailed coding catalog allows a three-level assessment of alexithymia development per item. Only non-representative norm data are available to serve as a guide. Reliability estimates of intraclass correlation correspond to 0.90 (p < 0.01) and reliability estimates to 0.88 (p < 0.01). A combined use of TAS and TSIA is suggested for effective assessment of alexithymia [46].

The short version of the Autism Spectrum Quotient (AQ-short) [47] consists of the three factors: interaction and spontaneity, imagination and creativity, and communication and reciprocity. The internal consistency of the factors ranged from 0.65 to 0.87, and the sensitivity analysis resulted in a cut-off value of 18.

The Beck Depression Inventory II (BDI-II) [48] is a self-report questionnaire with 21 multiple-choice questions. Cut-offs for BDI-II are as follows: 0–13 points no or minimal depressive symptoms, 14–19 points mild, 20–28 points moderate, 29–63 points severe depressive

symptoms. Retest reliability during one week is r = 0.93 with internal consistency in clinical and non-clinical samples of $0.84 \leq \alpha \leq 0.94$.

The Patient Health Questionnaire (PHQ-9) [49] is a nine-item component of the PHQ. Each item can be scored as 0 (not at all), 1 (on a single day), 2 (more than half of the days), or 3 (almost every day). Overall, the PHQ-9 score ranges from 0 to 27. Major depression can be diagnosed if any of the items indicate depressed mood and 5 or more items have a score of 2 or higher. Internal reliability was Cronbach's $\alpha$ = 0.89 in a representative primary care study.

The 20-item Prosopagnosia Index (PI-20) [50] is used to identify prosopagnosia traits. The index is a self-report instrument used to assess experience with face recognition. It is scored using a five-point scale (strongly agree to strongly disagree). The Cronbach's $\alpha$ of 0.96 shows a high internal consistency of the 20 items. Cut-off scores are 65–74 for mild, 75–84 for moderate and 85–100 for severe developmental prosopagnosia.

The Toronto Alexithymia Scale (TAS-20) [51] is a questionnaire that refers to people who tend to minimize emotional experience and focus attention externally and who have trouble describing and identifying emotions. The TAS-20 uses cut-off scoring $\leq 51$ = nonalexithymia, $\geq 61$ = alexithymia. Scores of 52 to 60 = possible alexithymia [52]. It is recommended to use the $33^{rd}$ percentile corresponding to $\leq 45$ (threshold for being surely nonalexithymic) and the $66^{th}$ percentile value corresponding to $\geq 52$ (threshold for being alexithymic) for experimental studies. To ensure correct group classification [53]. We were able to determine reliability coefficients Cronbach's $\alpha$ = .86 for the TAS-20 from the screening sample (N = 2924).

## 2.2 Participants

Subjects were recruited via posters and advertisements on social networks. The study procedure and data protection regulations were described in detail to the cted parties. Each subject received financial compensation of 25 Euro for expenses and signed an informed consent form. Subsequently, subjects accessed an online questionnaire [54] in which sociodemographic variables (age, gender, siblings, education), the PHQ-9, and the TAS-20 were collected and severe neurological or psychiatric disorders were queried. Exclusion criteria were insufficient knowledge of the German language, left-handedness, age under 18 or over 50 years, serious medical conditions such as endocrine disorders or coronary heart disease, use of psychotropic drugs, vigilance disorders, substance abuse, visual disorders, neurological disorders (including neuropathy and botulinum toxin use), or psychiatric disorders. The non-alexithymic control group studied here was characterized by a TAS-20 sum score <45 and originally included 38 participants. For technical reasons, 5 subjects were excluded from this study. Reasons for this were misplaced Affdex measurement points due to unfavorable lighting conditions or glasses although not every subject wearing glasses had to be excluded. Thus, 33 participants between the ages of 20 and 42 years (mean age = 25.24, SD = 5.73, SE = 0.99, 22 females, 11 males) were included. The clinically defined thresholds of AQ-short (cut-off value = 18), BDI-II (cut-off value = 13), PI20 (cut-off value = 65), PHQ-9 (cut-off value = 9) and TAS-20 (cut-off value = 51) were not exceeded by any of the subjects. The results of the subjects' psychometric tests are shown in Table 1.

## 2.3 Stimulus material

The stimulus material consisted of video sequences of adult faces showing the five basic affects (fear, joy, sadness, surprise, anger). Each video began with a neutral face that continuously built up a maximum affect expression (apex) over 2 seconds, which was presented for one second afterwards. Original portraits of adult individuals were taken from the Karolinska Directed Emotional Faces image set [55]. Deindividualized affect-expressive portraits for each

**Table 1. Descriptive statistics, n = 33.**

| Instrument | mean | sd | median | se |
|---|---|---|---|---|
| AQ-short | 4.64 | 2.29 | 4.00 | 0.40 |
| BDI-II | 2.00 | 2.21 | 2.00 | 0.38 |
| PI20 | 34.39 | 7.60 | 33.00 | 1.32 |
| PHQ-9 | 2.73 | 2.00 | 3.00 | 0.35 |
| TAS-20 | 31.94 | 5.34 | 31.00 | 0.93 |

AQ-short, Autism-Quotient short version sum score, cut-off value = 18; BDI-II, Beck Depression Inventory II sum score, cut-off value = 13; PHQ-9, 9-item depression module of the Patient Health Questionnaire sum score, cut-off value = 9; PI20, 20 item prosopagnosia index sum score, cut-off value = 65; sd, standard deviation; se, standard error; TAS-20, 20-item Toronto Alexithymia Scale sum score, cut-off value = 51.

gender and affect (five basic affects and neutral) were developed from the most valid portraits for each affect category [56]. This was realized by a digital overlay of the individual faces and resulted in affect prototypical facial patterns of basic affects in a purified way. These averaged affect prototypical portraits served as visual source material for the creation of video sequences of each basic affect and gender. For this reason, a software based morphing algorithm was used, which generated a naturalistic affect enrichment by interpolating video frames from neutral to maximal affect expression within 2000 ms. The final videos show dynamic sequences of naturalistic sliding facial affect amplification (2000 ms), followed by a static presentation of the apex of each basic affect (1000 ms). Both the averaged portraits and the dynamic video sequences were created and edited by using the software package Abrasoft Fantamorph Deluxe 5. The whole process of stimulus development and the proof of validity of the dynamic stimulus material was demonstrated by Müller et al. [57]. Here, specific mimic responses could be detected for each basic affect.

## 2.4 Procedure

The study was approved by the Ethics Committee of the Medical Faculty of Heinrich Heine University under the registration number 2016116024. Subjects were recruited via posters and advertisements on social networks. The study procedure and data protection regulations were described in detail to the interested parties. Each subject received financial compensation of 25 Euro for expenses and signed an informed consent form. Before starting the experiment, all participants had to pass simple functional tests for checking the reactivity and function of the facial nerve and visual perceptual ability. Subsequently, subjects completed the various psychometric instruments and clinical interviews (TAS-20, BDI-II, SCID, TSIA, PI20, and AQ-short). Only participants whose test scores were below the defined clinical threshold were admitted to the study. At the beginning of the experiment, subjects were shown the investigation cabin and it was explained that affect-expressive faces would be presented as videos and "bodily signals" would be measured simultaneously. To this end, participants were told to watch the videos attentively and empathize with the affects shown without imitating them. The texts and images were presented on a 24-inch TFT screen (AMW) with a resolution of 1920 x 1080 (60 Hz), at a distance of 1 m. Coordination of the experiment and presentation were controlled using PsychoPy v1.82.01 software [58]. The EMG activity was measured bipolar with Ag/AgCl miniature electrodes (Easy Cap E220N-CS-120) according to the guidelines of Fridlund and Cacioppo [59]. The electrodes were filled with electrolyte paste and attached to the left and right zygomatic and corrugator muscle regions. In addition, two reference electrodes were attached to the mastoids, and two additional electrodes were attached to the

temporal bone region for measurement of the electrooculogram (for later correction of artifacts). To ensure impedances below 10 k [59], the skin of the subjects was cleaned with alcohol and rubbed with an abrasive electrode paste before attaching the electrodes. After these procedures, the experiment was started. The stimulus material was presented for 3 seconds as described above (videos of affect-expressive faces, 2 seconds of affect enrichment, 1 second of apex). A black fixation cross on a white background was presented for a mean inter-stimulus–interval time of 5 seconds before each video presentation. The videos were presented in randomized order. Each subject watched 40 videos (five affects, two age groups, two genders, two runs). Subjects were filmed throughout the procedure to enable offline Affdex measurement and to monitor their cooperation in following the instructions and their compliance (vigilance, attention, involvement). The filming was performed with a digital camera, which took frontal video recordings of the subjects at a distance of 1 m. The resulting videos were first stored locally and later imported into the iMotions software and analyzed in iMotions using Affdex software.

## 2.5 Measurement of facial EMG

EMG Data were acquired from both sides of the face (left and right) from each muscle (zygomaticus and corrugator muscle). EMG activity during stimulus presentation was measured digitally with a sampling rate of 2000 Hz (digital polygraph EEG 1100 G; Nihon Kohden). The EMG signal was further processed offline using the Brain Vision Analyzer. A high-pass filter at 10 Hz and a low-pass filter at 1000 Hz were used. A notch filter (50 Hz) was also used to reduce electromagnetic interference. Before the start of the measurements, the subjects were asked to grimace in order to check the function of the measuring chain based on the initial EMG signals. The recorded signals were stored on a hard disk for further offline analyses and parametrization. Two independent reviewers checked the EMG measurements for artifacts (e.g. subject movement, electrode movements, current voltage drifts). Subsequently, the EMG signal was rectified and integrated stepwise for each 200 ms interval over 5000 ms. For better comparability with the interstimulus interval, 1 s before stimulus presentation was included in the analysis. Due to the dynamic affect buildup during the first 2 s of the stimulus presentation and the expected delayed facial response, an additional 1 s after stimulus presentation was evaluated. For subsequent analysis, a total of 25 200 ms intervals were used, i.e., 1 s before stimulus presentation, 3 s during stimulus presentation, and 1 s after stimulus presentation were each included in the measurement. EMG activity was determined baseline-corrected. The baseline was defined as the mean of the last 1000 ms before stimulus presentation. The preprocessed EMG data were imported into the statistical software package R for further analysis.

## 2.6 Measurement with Affdex

Affdex is a software program for automatic recognition of facial expressions based on the Facial Action Coding System [29]. First, the Viola-Jones algorithm [60] is used to recognize faces and mark the area relevant to the facial expression with a rectangular frame. Within this frame, 34 relevant measurement points on the face are identified and marked, and histogram-of-oriented-gradient features are obtained from the relevant measurement area. Using support-vector-machine classifiers trained with 10000 manually coded facial expressions, percentile ranks are obtained for each facial expression-relevant motion. Subsequently, affect-expressive facial expressions are inferred from the combination of different facial movements using the Emotional Facial Action Coding System and percentile ranks are also determined for the occurrence of one of the basic affects (anger, disgust, fear, joy, sadness, surprise, contempt) [61]. The FACS action units lip corner puller (action unit 12 according to FACS) and brow

lowerer (action unit 4 according to FACS) examined in this study were renamed Smile and brow furrow by Affectiva. However, we continue to use the official FACS nomenclature in this paper.

During the EMG measurement, subjects' faces were filmed with a video camera (C920 HD Pro Webcam), in a resolution of 1920x1080 (30 frames per second), which was located above the presentation screen, providing frontal footage of the subjects throughout the experiment. Because the camera was turned on a few minutes before the experiment began, the videos were initially trimmed to the actual length of the experiment. During initial trial measurements with Affdex, it was noticed that measurement points in the eyebrow area partially jumped into the area of the EMG electrodes that were responsible for the measurements of the corrugator muscle. To avoid erroneous measurements, the electrodes were retouched using video editing software (DaVinci Resolve, Blackmagic design), in consultation with iMotions technical support. For this purpose, the electrodes were covered with skin-colored areas and tracked over the entire course of the video in every single frame (framerate: 30 FPS). As a result the covers reliably covered only the area of the electrodes during facial movements of the subjects, but did not affect any areas relevant for measurement.

The resulting videos were then imported into the iMotions software (iMotions version 7.2). Within iMotions, markers were now added to the time segments in which the stimulus presentations took place, which enabled an assignment to the affect-expressive stimuli. The automatic marker import of iMotions often led to an inaccurate placement of the markers. Therefore, the markers were placed manually. To ensure that the markers were placed at the correct times, there was a small red light behind the subjects that was controlled by PsychoPy and turned off each time a stimulus presentation began. The correct order of the stimulus presentation could be viewed in PsychoPy. "Postprocessing" by the Affdex algorithm and subsequent data export now took place. The resulting data sets were imported into R (Version 4.1.0) for further parameterization and analysis.

## 2.7 Data reduction and analysis

Rectified individual EMG data were integrated for each 200 ms interval over 5000 ms (1000 ms before stimulus onset and 1000 ms after stimulus termination). Integrals were then averaged for both sides of the face left and right, for female and male stimuli and for first and second measurement, resulting in 25 x 200 ms averaged EMG integrals for each affect and subject.

The output of the Affdex data was in 40 ms intervals. These were first averaged over 200 ms. Subsequently, the data were averaged according to the EMG data for female and male stimulus material and for the first and the second measurement. For the correlation calculations, first and second measurement were not averaged. No distinction was made between the left and right half of the face by Affdex. Measurements of response to children stimuli were excluded.

Spearman correlations were then calculated between EMG activity and Affdex measurements at each measurement time point. For the presentation of the joy stimulus, the correlations between the zygomatic muscle and the lip corner puller action unit and the joy probability were calculated. For the presentation of the anger stimulus, the correlations between the corrugator muscle and the brow lowerer action unit and the anger probability were calculated. The correlation probabilities were tested for significance, and the significance level was set at $\alpha = 0.05$. Because of repeated measures, Hochberg-Benjamini corrections were applied for p values $\leq 0.05$. The Affdex data, the EMG data, and the respective correlations were plotted graphically together in Figs 1–4. Measurement time points at which Affdex and

EMG were significantly correlated (α≤0.05) were marked with an *. In addition, cross correlations were performed between the following time series: Zygomaticus muscle (EMG)- lip corner puller (Affdex); Zygomaticus muscle (EMG)- joy (Affdex); Corrugator muscle (EMG)- brow lowerer (Affdex); Corrugator muscle (EMG)- anger (Affdex).

## 3 Results

Fig 1 shows the facial mimicry response of the observer represented by the course of the EMG activity of the zygomaticus muscle and the probability of the lip corner puller action unit calculated by Affdex during the measurement interval when the video was displaying joy. In addition, the course of the Spearman correlation between EMG and Affdex at each measurement time point is shown. Fig 2 shows the course of the EMG activity of the corrugator muscle and the probability of the brow lowerer action unit calculated by Affdex. In addition, the course of the Spearman correlation between the two values is shown. Starting with the neutral face and extending to the apex, EMG activity for both affects is congruent with the affect enhancement of the stimuli. Approximately 400 ms after stimulus onset, EMG activity increases in parallel with the increasing affect expression of the stimulus. It increases to its maximum after 2000 ms, which corresponds to the arising apex of affect expression in the stimulus videos. In addition to the increase in mean values, an increase in variance is also evident. As expected, presentation of the joy stimulus led to an increase in zygomaticus muscle activity and presentation of the anger stimulus led to an increase in corrugator activity. The curve of the zygomaticus muscle reaches its maximum at a value of approx. 1800 μVx200 ms and drops to approx.

**Fig 1. Stimulus joy, measurement of zygomaticus muscle (EMG) and lip corner puller (Affdex).** Electromyographical activity [μV integrated over 25 x 200 ms interval (μV x 200 ms) +/- standard error] of zygomaticus muscle (blue line) and Affdex measurement (% +/- standard error) for the activity of the lip corner puller action unit (red line) in response to video clips of affect expressing faces of adults for the affect joy, whiskers represent the standard error, black line represents the Spearman correlation between electromyographical activity and Affdex at each measurement point, the symbol * indicates p ≤ 0.05.

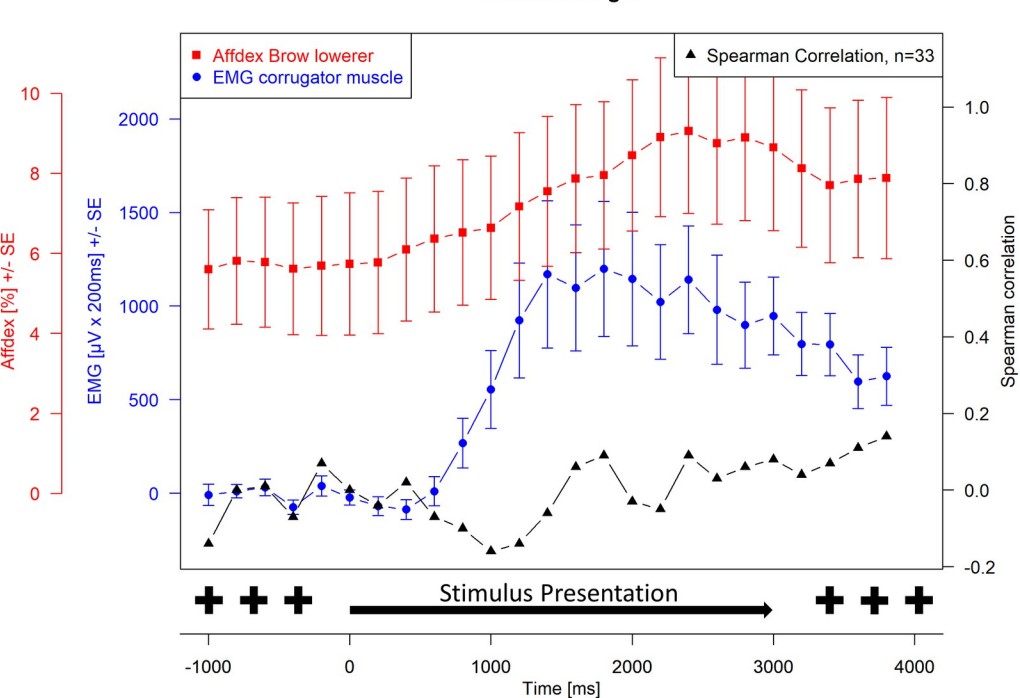

**Fig 2. Stimulus anger, measurement of corrugator muscle (EMG) and brow lowerer action unit (Affdex).**
Electromyographical activity [μV integrated over 25 x 200 ms interval (μV x 200 ms) +/- standard error] of corrugator muscle (blue line) and Affdex measurement (% +/- standard error) for the brow lowerer actionunit (red line) in response to video clips of affect expressing faces of adults for the affect anger, whiskers represent the standard error, black line represents the Spearman correlation between electromyographical activity and Affdex at each measurement point, the symbol * indicates p ≤ 0.05.

1200 μVx200 ms at the end of the measurement interval. The curve of corrugator muscle rises to a maximum value of 1300 μVx200 ms and drops to a value of approx. 500 μVx200 ms by the end of the measurement interval.

Table 2 contains the results of Spearman correlation calculations between zygomaticus muscle and lip corner puller including alpha error corrected p-values. Table 3 contains the results of Spearman correlation calculations between corrugator muscle and brow lowerer including alpha error corrected p-values.

The activities determined by Affdex for lip corner puller and brow lowerer, respectively, run differently. The activity of the action unit lip corner puller starts to increase 1200 ms after stimulus onset. The activity increases over a period of 2000 ms to its maximum of 6% (3000 ms after stimulus onset and 1000 ms after stimulus apex). As with the EMG measurement, the variance also increases in addition to the mean values. Thus, both the EMG of the zygomaticus muscle and the Affdex measurement of the lip corner puller action unit show an increase in activity upon presentation of the joy stimulus. However, lip corner puller activity proceeds with a latency of 800 ms compared with the course of the stimulus material and EMG activity.

Brow lowerer activity calculated by Affdex shows an increase from 6% during stimulus onset to its maximum of 8% approximately 2500 ms after stimulus onset. Unlike EMG activity, there is no change in variance, suggesting that this is not a stimulus-associated increase in brow lowerer activity calculated by Affdex but a random fluctuation.

The calculation of the Spearman correlation between the EMG activity for the zygomaticus muscle and the activity of the lip corner puller action unit calculated by Affdex for the affect

**Table 2. Stimulus joy, Spearman correlation between zygomaticus muscle (EMG) and lip corner puller (Affdex).**

| Time [ms] | n-value | r-value | Effect size | df | lower ci | upper ci | p-value | Corrected p-value |
|---|---|---|---|---|---|---|---|---|
| -1000 | 66 | 0.1389497 | 0.01930702 | 64 | -0.1066704 | 0.3685868 | 0.2658492 | - |
| -800 | 66 | 0.03903398 | 0.001523652 | 64 | -0.2049349 | 0.2784362 | 0.7556689 | - |
| -600 | 66 | -0.1423648 | 0.02026774 | 64 | -0.3715936 | 0.1032248 | 0.254162 | - |
| -400 | 66 | -0.0714157 | 0.005100206 | 64 | -0.3081226 | 0.173618 | 0.568795 | - |
| -200 | 66 | -0.1102954 | 0.01216508 | 64 | -0.3431673 | 0.1353505 | 0.3779834 | - |
| 0 | 66 | -0.1694939 | 0.02872818 | 64 | -0.3953098 | 0.07564198 | 0.1736616 | - |
| 200 | 66 | -0.1941923 | 0.03771065 | 64 | -0.4166425 | 0.0501999 | 0.1181946 | - |
| 400 | 66 | -0.0209880 | 0.0004404974 | 64 | -0.2616915 | 0.2221733 | 0.8671591 | - |
| 600 | 66 | 0.1098222 | 0.01206092 | 64 | -0.1358208 | 0.3427446 | 0.3800456 | - |
| 800 | 66 | 0.1252305 | 0.01568268 | 64 | -0.1204532 | 0.356459 | 0.316397 | - |
| 1000 | 66 | 0.1943012 | 0.03775296 | 64 | -0.0500870 | 0.416736 | 0.1179845 | - |
| 1200 | 66 | 0.2110547 | 0.04454409 | 64 | -0.0326456 | 0.4310676 | 0.0889287 | - |
| 1400 | 66 | 0.1922236 | 0.03694991 | 64 | -0.0522396 | 0.414951 | 0.1220435 | - |
| 1600 | 66 | 0.2111252 | 0.04457385 | 64 | -0.0325719 | 0.4311277 | 0.0888196 | - |
| 1800 | 66 | 0.2271216 | 0.05158422 | 64 | -0.0157785 | 0.4447084 | 0.0666633 | - |
| 2000 | 66 | 0.3006293 | 0.09037798 | 64 | 0.06319469 | 0.505855 | 0.01417941 | 0.3544852 |
| 2200 | 66 | 0.4328895 | 0.1873933 | 64 | 0.2131938 | 0.6109147 | 0.00028311 | 0.00707775 |
| 2400 | 66 | 0.4593286 | 0.2109828 | 64 | 0.2444747 | 0.6311903 | 0.00010461 | 0.00261525 |
| 2600 | 66 | 0.542301 | 0.2940904 | 64 | 0.3456343 | 0.6933309 | $2.56027e^{-06}$ | $6.400675e^{-05}$ |
| 2800 | 66 | 0.5338247 | 0.2849688 | 64 | 0.3350861 | 0.6870841 | $3.91689e^{-06}$ | $9.792225e^{-05}$ |
| 3000 | 66 | 0.5420904 | 0.293862 | 64 | 0.3453717 | 0.693176 | $2.58782e^{-06}$ | $6.46955e^{-05}$ |
| 3200 | 66 | 0.467337 | 0.2184039 | 64 | 0.2540387 | 0.6372858 | $7.61925e^{-06}$ | 0.0001904812 |
| 3400 | 66 | 0.460306 | 0.2118816 | 64 | 0.2456397 | 0.6319354 | 0.00010075 | 0.00251875 |
| 3600 | 66 | 0.4806718 | 0.2310454 | 64 | 0.270057 | 0.6473885 | $4.40978e^{-05}$ | 0.001102445 |
| 3800 | 66 | 0.4313402 | 0.1860544 | 64 | 0.2113746 | 0.6097192 | 0.00029936 | 0.007484 |

joy shows an increase in Spearman correlation shortly after the onset of stimulus presentation, but without becoming significant. The Spearman correlation coefficients become significant ($p \leq 0.05$) from 2200 ms after the onset of stimulus presentation and increase to a maximum of approximately 0.5 3000 ms after stimulus presentation. Zygomaticus muscle activity and lip corner puller action unit activity calculated by Affdex correlate significantly with each other until the end of the measurement interval 4000 ms after stimulus presentation.

Calculation of Spearman correlation between EMG activities for corrugator muscle and brow lowerer action unit calculated by Affdex for affect anger shows no relevant increase in Spearman correlation. Corrugator muscle activity does not significantly correlate with brow lowerer action unit activity calculated by Affdex at any time point

Fig 3 shows again the course of the EMG activity of the zygomaticus muscle and the probability of joy calculated by Affdex. In addition, the course of the Spearman correlation between the two values is shown. Fig 4 shows the course of the EMG activity of the corrugator muscle and the probability of anger calculated by Affdex. The probability for the affect joy calculated by Affdex shows an increase 1400 ms after stimulus onset from 0% to 2.5%. The curve reaches its maximum 500 ms after the apex of the affect expression of the stimulus (2500 ms after stimulus onset). As with the EMG measurement and the action unit measurement, the variance increases here in addition to the mean values. Thus, both the EMG of the zygomaticus muscle and the Affdex measure of joy probability show an increase upon presentation of the joy stimulus. However, the joy probability calculated by Affdex progresses with a latency of 1000 ms compared with the progress of the stimulus material and EMG activity.

**Table 3. Stimulus anger, Spearman correlation between corrugator muscle (EMG) and brow lowerer action unit (Affdex).**

| Time [ms] | n-value | r-value | Effect size | df | lower ci | upper ci | p-value |
|---|---|---|---|---|---|---|---|
| -1000 | 66 | -0.1437431 | 0.02066208 | 64 | -0.3728058 | 0.1018325 | 0.2495456 |
| -800 | 66 | 0.001796796 | $3.228476e^{-06}$ | 64 | -0.2403405 | 0.2437236 | 0.988576 |
| -600 | 66 | 0.01142858 | 0.0001306124 | 64 | -0.2312438 | 0.2527622 | 0.9274329 |
| -400 | 66 | -0.06701946 | 0.004491608 | 64 | -0.3041192 | 0.177899 | 0.5928798 |
| -200 | 66 | 0.07088798 | 0.005025106 | 64 | -0.1741324 | 0.3076425 | 0.5716619 |
| 0 | 66 | -0.00206789 | $4.276153e^{-06}$ | 64 | -0.2439785 | 0.2400851 | 0.9868526 |
| 200 | 66 | -0.04331385 | 0.00187609 | 64 | -0.2823863 | 0.2008242 | 0.7298522 |
| 400 | 66 | 0.01619883 | 0.0002624021 | 64 | -0.2267228 | 0.2572231 | 0.8972833 |
| 600 | 66 | -0.06906 | 0.004769284 | 64 | -0.3059784 | 0.1759131 | 0.5816441 |
| 800 | 66 | -0.09947335 | 0.009894947 | 64 | -0.3334774 | 0.1460763 | 0.4268122 |
| 1000 | 66 | -0.1558416 | 0.0242866 | 64 | -0.3834125 | 0.08956966 | 0.2114737 |
| 1200 | 66 | -0.1362269 | 0.01855777 | 64 | -0.366186 | 0.1094133 | 0.275422 |
| 1400 | 66 | -0.05674876 | 0.003220422 | 64 | -0.2947333 | 0.1878643 | 0.6508473 |
| 1600 | 66 | 0.06390056 | 0.004083282 | 64 | -0.1809305 | 0.3012738 | 0.6102384 |
| 1800 | 66 | 0.08782626 | 0.007713452 | 64 | -0.1575557 | 0.3229932 | 0.4831607 |
| 2000 | 66 | -0.03159638 | 0.0009983312 | 64 | -0.2715525 | 0.2120581 | 0.8011581 |
| 2200 | 66 | -0.04826307 | 0.002329324 | 64 | -0.286944 | 0.1960599 | 0.7003679 |
| 2400 | 66 | 0.08563286 | 0.007332987 | 64 | -0.1597101 | 0.3210123 | 0.4941963 |
| 2600 | 66 | 0.03358398 | 0.001127884 | 64 | -0.210157 | 0.2733945 | 0.7889302 |
| 2800 | 66 | 0.05541565 | 0.003070894 | 64 | -0.1891541 | 0.2935117 | 0.6585348 |
| 3000 | 66 | 0.07864943 | 0.006185733 | 64 | -0.1665538 | 0.3146918 | 0.5301898 |
| 3200 | 66 | 0.04100801 | 0.001681657 | 64 | -0.20304 | 0.2802591 | 0.7437263 |
| 3400 | 66 | 0.07333763 | 0.005378408 | 64 | -0.1717436 | 0.3098702 | 0.5584113 |
| 3600 | 66 | 0.1148427 | 0.01318885 | 64 | -0.1308265 | 0.3472241 | 0.3585125 |
| 3800 | 66 | 0.0182875 | 0.1352313 | 64 | -0.1104154 | 0.3653074 | 0.2789792 |

n-value, Number of cases per correlation, r-value, correlation coefficient, df, degrees of freedom, lower ci, lower confidence interval, upper ci, upper confidence interval

Table 4 contains the results of Spearman correlation calculations between zygomaticus muscle and Joy including alpha error corrected p-values. Table 5 contains the results of Spearman correlation calculations between corrugator muscle and anger including alpha error corrected p-values.

The probability for affect anger calculated by Affdex shows no increase and remains at 0% throughout the stimulus presentation.

The calculation of Spearman correlation coefficients between EMG activity of the zygomaticus muscle and the probability of detecting joy calculated by Affdex during the presentation of the joy video shows a value of -0.2 until it increases to 0 1000 ms after stimulus onset and increases from 1600 ms to a maximum value of 0.4, 2600 ms after stimulus onset. The Spearman correlation coefficients become significant ($p \leq 0.05$) at 3400 ms, 3800 ms and 4000 ms.

Calculation of Spearman correlation between EMG activities for corrugator muscle and anger probability for the anger affect calculated by Affdex shows an increase of Spearman correlation to nearly 0.2 from 3800 ms after stimulus presentation.

Fig 5 shows the results of the cross-correlation calculation between the EMG activity of the zygomaticus muscle and the lip corner puller activity determined by Affdex during the presentation of the joy stimulus. the largest cross-correlation coefficient was found at a lag of 5 and was 0.256, indicating lagged matching in temporal patterns between EMG and Affdex.

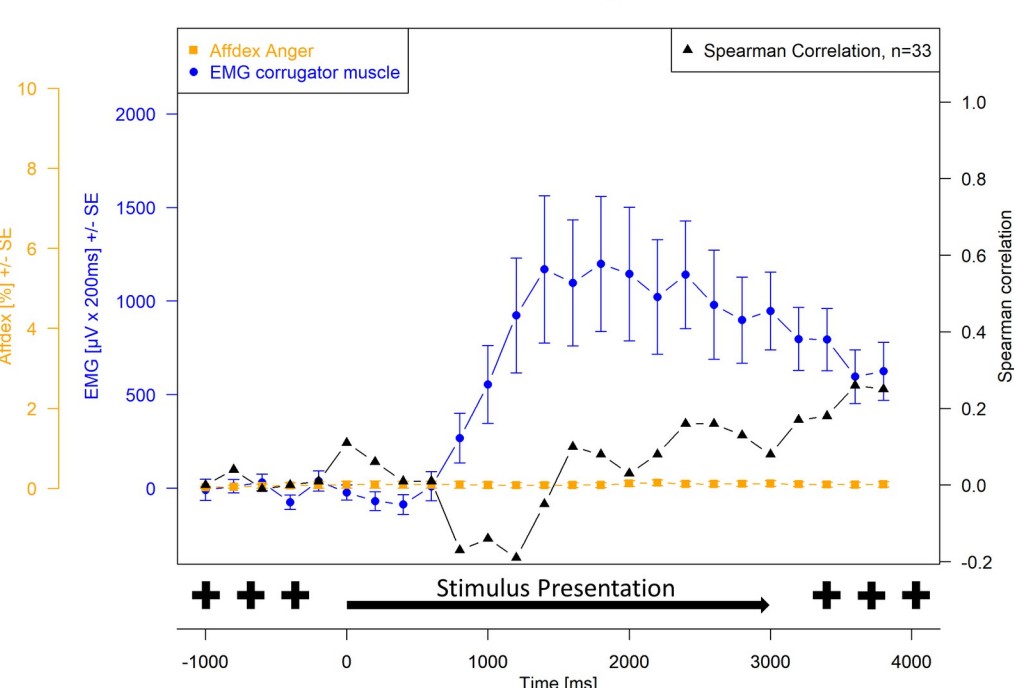

**Fig 4. Stimulus anger, measurement of corrugator muscle (EMG) and affect anger (Affdex).** Electromyographical activity [μV integrated over 25 x 200 ms interval (μV x 200 ms) +/- standard error] of corrugator muscle (blue line) and Affdex measurement (% +/- standard error) for the affect anger (orange line) in response to video clips of affect expressing faces of adults for the affect anger, whiskers represent the standard error, black line represents the Spearman correlation between electromyographical activity and Affdex at each measurement point, the symbol * indicates p ≤ 0.05.

Table 6 contains the cross-correlation coefficients of all measurements shown so far. For the calculation of the cross-correlation coefficient between zygomaticus muscle and the affect joy during the presentation of the joy stimulus, the largest value was also shown at a lag of 5. The cross-correlation coefficient here was 0.174. During the presentation of the anger stimulus, the cross-correlation coefficients show no relevant increase.

## 4 Discussion

This study is the first in which the facial mimicry response of healthy subjects to dynamic affect-enhancing videos (joy and anger) was measured simultaneously using EMG and Affdex to compare the suitability of these two measurement methods.

EMG measurement has been the gold standard for measuring facial mimicry. However, it requires a complex measuring apparatus and experience in the application and interpretation of EMG signals. Affdex promises a time-saving and easy-to-interpret analysis of facial expressions. In addition, measurement electrodes in the face would not be necessary. This could open up a wide range of applications in affect research.

First, the EMG activity of zygomaticus muscle and corrugator muscle was directly compared to each other using the Affdex action units lip corner puller and brow lowerer, which are based on FACS [43]. This allows for a direct comparison of measurement sensitivity, as these action units represent the visible correlates to the underlying target muscles [29]. Second, EMG activity was compared to affect probabilities measured by Affdex, as Affdex uses other

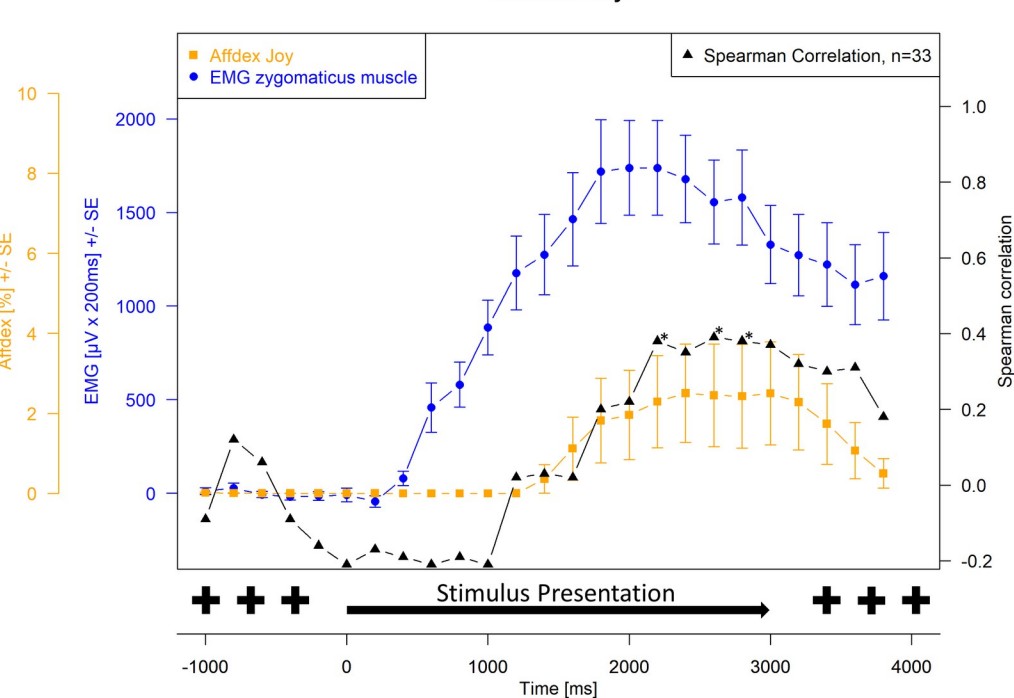

**Fig 3. Stimulus joy, measurement of zygomaticus muscle (EMG) and affect joy (Affdex).** Electromyographical activity [μV integrated over 25 x 200 ms interval (μV x 200 ms) +/- standard error] of zygomaticus muscle (blue line) and Affdex measurement (% +/- standard error) for the affect joy (orange line) in response to video clips of affect expressing faces of adults for the affect joy, whiskers represent the standard error, black line represents the Spearman correlation between electromyographical activity and Affdex at each measurement point, the symbol * indicates p ≤ 0.05.

data for measurement in addition to the lip corner puller and brow lowerer action units according to EMFACS [39].

We expected comparable results for Affdex and EMG measurements [41, 42]. However, there was also evidence for reduced measurement performance of Affdex for subtle affect expressions, as expected for facial mimicry [40].

In healthy subjects, it has been shown that facial mimicry could be induced by affective stimulus material. The muscle activity of the zygomaticus and corrugator muscles measured by EMG reflected the valence of the presented affects joy and anger [33, 62].

The Affdex measurement for the lip corner puller action unit and the affect joy also showed an increase and a significant correlation with the EMG measurement of the zygomaticus muscle during the presentation of the joy stimulus. However, the rise of the Affdex trace for lip corner puller did not begin until 1200 ms after stimulus onset and approximately 800 ms after the rise of the EMG trace. Sato et al. [63] demonstrated that human FACS coders detect the facial mimicry response to a dynamic happy stimulus after 817 (±200) ms after stimulus onset. The Affdex trace for joy rose 200 ms later than the trace for lip corner puller. The Affdex trace for lip corner puller reached its maximum at an average of 6.02% and the Affdex trace for joy at an average of 2.51%. These values correspond to relatively low expressions. Kulke et al. [41] studied a healthy cohort who imitated faces with maximum affect expression. Here, Affdex measured a maximum mean of 69.56% for lip corner puller and a maximum mean of 67.53% for joy when imitating joy. Thus, it was shown that Affdex is generally capable of measuring the facial mimicry response to the joy stimulus, however it´s reactivity starts much later. The

**Table 4. Stimulus joy, Spearman correlation between zygomaticus muscle (EMG) and affect joy (Affdex).**

| Time [ms] | n-value | r-value | Effect size | df | lower ci | upper ci | p-value | Corrected p-value |
|---|---|---|---|---|---|---|---|---|
| -1000 | 66 | -0.0854732 | 0.007305663 | 64 | -0.320868 | 0.1598668 | 0.4950049 | - |
| -800 | 66 | 0.1178771 | 0.01389501 | 64 | -0.1278019 | 0.3499264 | 0.345869 | - |
| -600 | 66 | 0.06178529 | 0.003817422 | 64 | -0.1829838 | 0.2993417 | 0.6221349 | - |
| -400 | 66 | -0.0896203 | 0.008031798 | 64 | -0.3246119 | 0.1557918 | 0.4742325 | - |
| -200 | 66 | -0.1555527 | 0.02419664 | 64 | -0.38316 | 0.0898633 | 0.212332 | - |
| 0 | 66 | -0.2120662 | 0.04497207 | 64 | -0.4319294 | 0.03158785 | 0.08737338 | - |
| 200 | 66 | -0.1696811 | 0.02879168 | 64 | -0.3954725 | 0.07545029 | 0.1731798 | - |
| 400 | 66 | -0.1932284 | 0.03733721 | 64 | -0.4158145 | 0.05119882 | 0.1200672 | - |
| 600 | 66 | -0.2055995 | 0.04227115 | 64 | -0.4264132 | 0.03834118 | 0.09769558 | - |
| 800 | 66 | -0.0854732 | 0.007305663 | 64 | -0.320868 | 0.1598668 | 0.4950049 | - |
| 1000 | 66 | -0.2147373 | 0.04611211 | 64 | -0.434203 | 0.02879192 | 0.08336919 | - |
| 1200 | 66 | 0.01678515 | 0.0002817413 | 64 | -0.2261664 | 0.2577707 | 0.8935867 | - |
| 1400 | 66 | 0.03147738 | 0.0009908255 | 64 | -0.2121718 | 0.2714422 | 0.8018917 | - |
| 1600 | 66 | 0.02088638 | 0.0004362409 | 64 | -0.22227 | 0.2615967 | 0.8677967 | - |
| 1800 | 66 | 0.1979175 | 0.03917134 | 64 | -0.0463349 | 0.4198389 | 0.111167 | - |
| 2000 | 66 | 0.2154148 | 0.04640354 | 64 | -0.0280821 | 0.4347793 | 0.08237689 | - |
| 2200 | 66 | 0.3833971 | 0.1469933 | 64 | 0.155824 | 0.5723215 | 0.00148503 | 0.03712575 |
| 2400 | 66 | 0.3502786 | 0.1226951 | 64 | 0.1182729 | 0.5460204 | 0.00393588 | 0.098397 |
| 2600 | 66 | 0.3916563 | 0.1533947 | 64 | 0.1652921 | 0.5788205 | 0.00114594 | 0.0286485 |
| 2800 | 66 | 0.3758274 | 0.1412462 | 64 | 0.1471827 | 0.566344 | 0.00187239 | 0.04680975 |
| 3000 | 66 | 0.3703136 | 0.1371322 | 64 | 0.1409103 | 0.5619774 | 0.00220921 | 0.05523025 |
| 3200 | 66 | 0.3192721 | 0.1019347 | 64 | 0.08370778 | 0.5210417 | 0.00897530 | 0.2243825 |
| 3400 | 66 | 0.3043612 | 0.09263574 | 64 | 0.06728495 | 0.5089052 | 0.01296778 | 0.3241945 |
| 3600 | 66 | 0.3106878 | 0.09652691 | 64 | 0.07423745 | 0.5140647 | 0.01111744 | 0.277936 |
| 3800 | 66 | 0.03315688 | -0.062706 | 64 | 0.1820903 | 0.4062202 | 0.1433829 | - |

n-value, Number of cases per correlation, r-value, correlation coefficient, df, degrees of freedom, lower ci, lower confidence interval, upper ci, upper confidence interval, corrected p-value (Benjamini Hochberg correction for multiple testing)

Affdex measurement for the action unit brow lowerer showed no stimulus-associated change and no significant correlation with the EMG measurement for the corrugator muscle at any time during the measurement. The Affdex trace for affect anger showed no deflection during stimulus presentation. Higher levels are found in a healthy cohort that imitated anger stimuli [41]. Here, Affdex measured a maximum mean of 36.72% for brow lowerer and a maximum mean of 8.88% for anger when imitating anger. Affdex thus performs relatively poor in our trials measuring the facial mimicry response to the anger stimulus.

Since the measurements are time series, we also calculated cross-correlation correlations. These additional calculations provided statistical evidence about a lagged matching in temporal patterns between EMG and Affdex.

For the measurements of corrugator muscle, brow lowerer and anger, the cross-correlation coefficients show no relevant increase.

Similar studies already indicated low sensitivity of automated affect detection for subtle affect expressions [40, 42, 64]. However, hedonic affect could be measured better than anhedonic affect [65], which is consistent with the results of present study.

It remains unclear why Affdex detects the mimicry response for joy but not for anger, although the EMG measures muscle activity in both cases. Other studies also showed weaker recognition performance of Automated Facial Coding for anger compared to joy [40, 66]. One

**Table 5. Stimulus anger, Spearman correlation between corrugator muscle (EMG) and affect anger (Affdex).**

| Time [ms] | n-value | r-value | Effect size | df | lower ci | upper ci | p-value | Corrected p-value |
|---|---|---|---|---|---|---|---|---|
| -1000 | 66 | -0.0045819 | $2.099417e^{-05}$ | 64 | -0.2463415 | 0.2377145 | 0.9708735 | - |
| -800 | 66 | 0.04370906 | 0.001910482 | 64 | -0.2004442 | 0.2827506 | 0.7274827 | - |
| -600 | 66 | -0.0091734 | $8.415112e^{-05}$ | 64 | -0.2506497 | 0.2333775 | 0.9417245 | - |
| -400 | 66 | 0.004527421 | $2.049754e^{05}$ | 64 | -0.2377659 | 0.2462903 | 0.9712199 | - |
| -200 | 66 | 0.0121565 | 0.0001477805 | 64 | -0.2305546 | 0.2534436 | 0.9228245 | - |
| 0 | 66 | 0.1085402 | 0.01178098 | 64 | -0.1370941 | 0.341599 | 0.3856662 | - |
| 200 | 66 | 0.05565026 | 0.003096951 | 64 | -0.1889272 | 0.2937268 | 0.6571793 | - |
| 400 | 66 | 0.0129309 | 0.0001672082 | 64 | -0.2298211 | 0.2541682 | 0.9179247 | - |
| 600 | 66 | 0.006851618 | $4.694467e^{-05}$ | 64 | -0.2355718 | 0.2484723 | 0.9564572 | - |
| 800 | 66 | -0.1680629 | 0.02824514 | 64 | -0.3940664 | 0.07710626 | 0.1773758 | - |
| 1000 | 66 | -0.1378292 | 0.01899689 | 64 | -0.3675991 | 0.1077997 | 0.2697614 | - |
| 1200 | 66 | -0.1923704 | 0.03700637 | 64 | -0.4150772 | 0.05208755 | 0.1217531 | - |
| 1400 | 66 | -0.04926441 | 0.002426982 | 64 | -0.2878648 | 0.1950946 | 0.6944542 | - |
| 1600 | 66 | 0.0970532 | 0.009419324 | 64 | -0.1484671 | 0.3313036 | 0.4382027 | - |
| 1800 | 66 | 0.07548504 | 0.005697991 | 64 | -0.1696472 | 0.3118209 | 0.5469167 | - |
| 2000 | 66 | 0.0339959 | 0.001155721 | 64 | -0.2097628 | 0.273776 | 0.7864023 | - |
| 2200 | 66 | 0.07830402 | 0.00613152 | 64 | -0.1668917 | 0.3143786 | 0.5320032 | - |
| 2400 | 66 | 0.1555797 | 0.02420504 | 64 | -0.0898359 | 0.3831835 | 0.2122518 | - |
| 2600 | 66 | 0.1603847 | 0.02572325 | 64 | -0.0849456 | 0.38738 | 0.1982959 | - |
| 2800 | 66 | 0.1347437 | 0.01815586 | 64 | -0.110906 | 0.364877 | 0.2807322 | - |
| 3000 | 66 | 0.08484137 | 0.007198058 | 64 | -0.1604869 | 0.320297 | 0.4982106 | - |
| 3200 | 66 | 0.166375 | 0.02768064 | 64 | -0.0788322 | 0.3925986 | 0.1818309 | - |
| 3400 | 66 | 0.1750089 | 0.03062812 | 64 | -0.0699885 | 0.4000945 | 0.1598766 | - |
| 3600 | 66 | 0.2580593 | 0.0665946 | 64 | 0.01709419 | 0.4706931 | 0.0364382 | 0.910955 |
| 3800 | 66 | 0.2467782 | 0.06089948 | 64 | 0.00504685 | 0.4612606 | 0.0457665 | 1 |

n-value, Number of cases per correlation, r-value, correlation coefficient, df, degrees of freedom, lower ci, lower confidence interval, upper ci, upper confidence interval

reason for this could be that the joy stimulus leads to higher and longer-lasting muscle activity than the anger stimulus. While the zygomaticus muscle activity increases to about 1800 µV x 200 ms and remains at over 1200 µV x 200 ms until the end of the measurement interval, the corrugator muscle activity only increases to a maximum of about 1300 µV x 200 ms and decreases to about 500 µV x 200 ms until the end of the measurement interval. The course of EMG activity could provide clues to the activity of the action units measured by Affdex. It is possible that the muscle activity of the corrugator muscle is not high enough to activate the brow lowerer action unit to a sufficient extent to detect it measurable by Affdex. Other studies also showed a stronger EMG response to hedonic stimuli than to anhedonic stimuli [42]. Even if EMG activity can be reliably measured, quantitatively these are extremely small increases in activation in the EMG. It is conceivable that EMG can measure muscle activity that does not result in any visible change in the face.

Another explanation could be the simultaneous application of EMG by skin electrodes and Affdex measurement. Kulke et al. [41] found that when measuring imitated affect with Affdex and simultaneous EMG measurement, the measurement result was only slightly worsened by the EMG electrodes used on the face. We observed that the Affdex measurement points, which are regularly located at the eyebrows, jumped over longer time intervals to the EMG electrodes located at the forehead above the eyebrows. This occurred even though the electrodes did not cover the areas relevant to Affdex. Therefore, in this study, to achieve consistently functioning

## Stimulus Joy

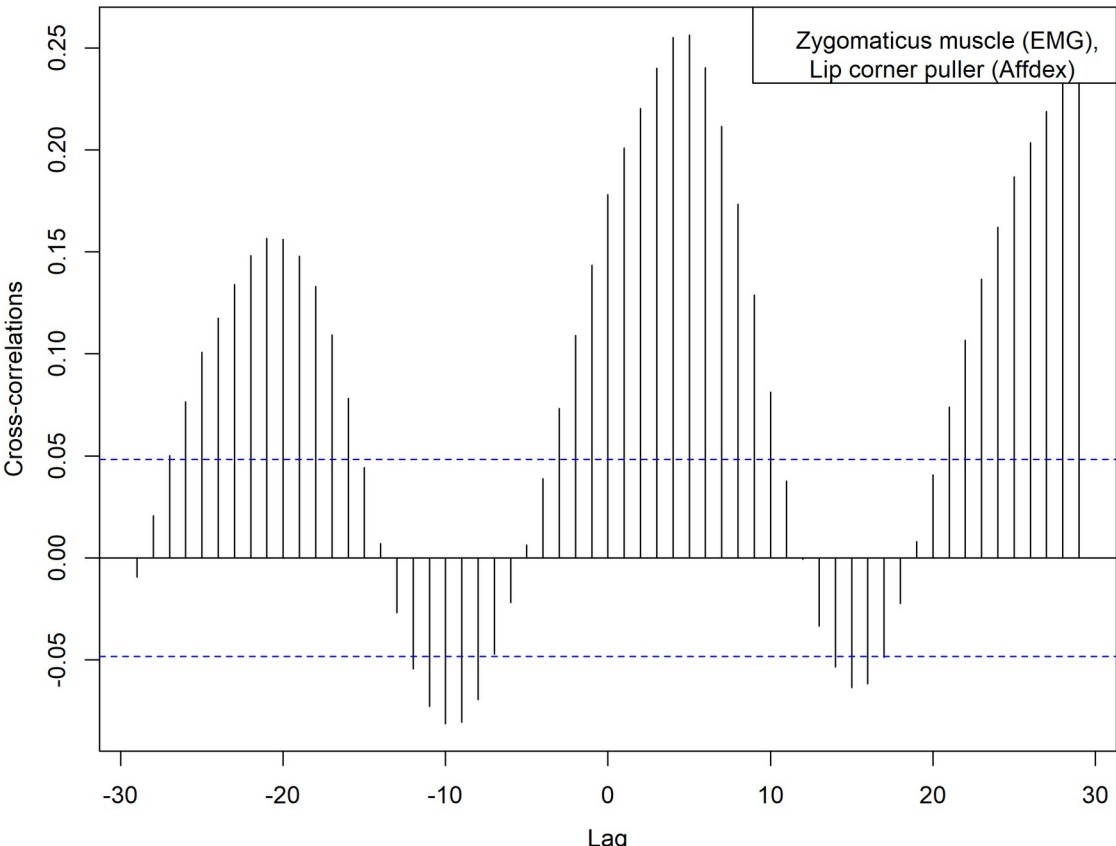

**Fig 5. Stimulus joy, cross-correlation coefficients between zygomaticus muscle (EMG) and lip corner puller (Affdex).** Cross-correlation coefficients between the EMG activity of the zygomaticus muscle and the lip corner puller activity determined by Affdex during the presentation of the joy stimulus (black vertical lines). The blue dashed line represents the 95% confidence interval.

Affdex measurements, the EMG electrodes in the video footage of the test subjects observing the video sequences were retouched after consultation with the iMotions support team. After retouching, the measurement points were consistently located on the eyebrows, so that the Affdex measurement may no longer have been affected. Of course, it is conceivable that the retouching obscured needed cues, and thus impaired Affdex' anger and brow lowerer detection in particular. In this case, this study is not suitable to assess Affdex's performance for measuring brow lowerer activity. The measurement points responsible for measuring Affdex' lip corner puller action unit and joy were in the correct positions throughout the measurement interval.

In conducting this study, in addition to the above mentioned advantages of Affdex compared to EMG disadvantages of Affdex were also noticed. A total of 5 subjects were excluded due to incorrect measurements by Affdex caused by unfavorable lighting conditions or glasses. To prevent further measurement errors, EMG electrodes had to be retouched as described above, which was technically challenging and very time-consuming. We performed the Affdex measurements on post-processed videos. The import of stimulus markers provided by iMotions for this procedure sometimes resulted in temporally offset markers. As a result, the markers had to be inserted manually to accurately mark the times at which stimuli were presented. This procedure was also very time-consuming.

**Table 6. Cross-correlation coefficients by lag.**

| Stimulus joy, cross-correlation coefficients between zygomaticus muscle (EMG) and lip corner puller (Affdex) by lag | | | | | | | | | | | | |
|---|---|---|---|---|---|---|---|---|---|---|---|---|
| 0 | 1 | 2 | 3 | 4 | 5 | 6 | 7 | 8 | 9 | 10 | 11 | 12 |
| 0.178 | 0.201 | 0.220 | 0.240 | 0.255 | 0.256 | 0.240 | 0.211 | 0.173 | 0.129 | 0.081 | 0.038 | -0.001 |
| 13 | 14 | 15 | 16 | 17 | 18 | 19 | 20 | 21 | 22 | 23 | 24 | 25 |
| -0.033 | -0.053 | -0.064 | -0.062 | -0.049 | -0.022 | 0.008 | 0.041 | 0.074 | 0.107 | 0.137 | 0.162 | 0.187 |
| Stimulus anger, cross-correlation coefficients between corrugator muscle (EMG) and brow lowerer (Affdex) by lag | | | | | | | | | | | | |
| 0 | 1 | 2 | 3 | 4 | 5 | 6 | 7 | 8 | 9 | 10 | 11 | 12 |
| 0.038 | 0.045 | 0.051 | 0.055 | 0.058 | 0.055 | 0.049 | 0.039 | 0.033 | 0.023 | 0.013 | 0.006 | 0.003 |
| 13 | 14 | 15 | 16 | 17 | 18 | 19 | 20 | 21 | 22 | 23 | 24 | 25 |
| 0.007 | 0.013 | 0.019 | 0.025 | 0.031 | 0.039 | 0.045 | 0.050 | 0.058 | 0.067 | 0.072 | 0.079 | 0.085 |
| Stimulus joy, cross-correlation coefficients between zygomaticus muscle (EMG) and joy (Affdex) by lag | | | | | | | | | | | | |
| 0 | 1 | 2 | 3 | 4 | 5 | 6 | 7 | 8 | 9 | 10 | 11 | 12 |
| 0.114 | 0.128 | 0.143 | 0.157 | 0.170 | 0.174 | 0.161 | 0.135 | 0.103 | 0.073 | 0.045 | 0.019 | -0.004 |
| 13 | 14 | 15 | 16 | 17 | 18 | 19 | 20 | 21 | 22 | 23 | 24 | 25 |
| -0.024 | -0.038 | -0.047 | -0.044 | -0.029 | -0.010 | 0.008 | 0.029 | 0.051 | 0.074 | 0.096 | 0.116 | 0.132 |
| Stimulus anger, cross-correlation coefficients between corrugator muscle (EMG) and anger (Affdex) by lag | | | | | | | | | | | | |
| 0 | 1 | 2 | 3 | 4 | 5 | 6 | 7 | 8 | 9 | 10 | 11 | 12 |
| -0.004 | -0.001 | 0.003 | 0.007 | 0.011 | 0.014 | 0.014 | 0.009 | 0.005 | 0.001 | -0.001 | -0.004 | -0.006 |
| 13 | 14 | 15 | 16 | 17 | 18 | 19 | 20 | 21 | 22 | 23 | 24 | 25 |
| -0.007 | -0.007 | -0.005 | -0.004 | 0.000 | 0.007 | 0.013 | 0.026 | 0.038 | 0.037 | 0.038 | 0.042 | 0.047 |

Numbers 0–25, lags, EMG, Electromyography

Technical improvements could resolve these problems and significantly improve the application.

As mentioned earlier, retouching the electrodes was time consuming. Future studies should either not use electrodes when measuring simultaneously with Affdex or place them on the face and cover them up so that they do not interfere with Affdex measurement.

Future studies could additionally check the subject videos with human FACS raters. This type of validity check would be very time-consuming but would clarify whether Affdex does not detect changes in mimic musculature that are visible to humans.

The present study focused on measuring the facial mimicry of the most commonly studied affects, joy and anger. Future studies could investigate other affects such as fear, disgust, sadness, and surprise.

## 4.1 Conclusion

The present study demonstrates that Affdex can measure a facial mimicry response for the affect joy. Despite the delayed measurement compared to the established EMG measurement, Affdex shows a valid performance. Nevertheless, it still does not match the highly sensitive EMG and therefore needs further improvement for measuring subtle affect expressions. It remains unclear how well Affdex detects the facial mimicry response to an angry stimulus, because in this study the electrodes measuring the corrugator muscle probably confounded Affdex. Should the measurement performance of Affdex improve significantly in the future and enable the measurement of subtle affect expressions, it could develop into a promising measurement instrument with a broad range of applications. Especially naturalistic experimental settings that require non-contact measurement of affective responses could benefit from Affdex. However, EMG has been superior in capturing the temporal and dynamic course characteristics of affect-expressive mimicry, at least for the basic affects studied here. EMG thus remains the gold standard for measuring facial mimicry.

## Acknowledgments

We would like to thank Lotte Wagner-Douglas, Claudius Rehagel and Alexandra Schwatlo for help with data collection and data analysis.

## Author Contributions

**Conceptualization:** Jan-Frederik Westermann, Marc Nordmann, Matthias Franz.

**Data curation:** Jan-Frederik Westermann, Ralf Schäfer, Marc Nordmann, Peter Richter.

**Formal analysis:** Jan-Frederik Westermann, Peter Richter.

**Investigation:** Jan-Frederik Westermann.

**Methodology:** Ralf Schäfer, Tobias Müller, Matthias Franz.

**Project administration:** Matthias Franz.

**Resources:** Matthias Franz.

**Software:** Jan-Frederik Westermann.

**Supervision:** Ralf Schäfer, Matthias Franz.

**Validation:** Ralf Schäfer, Matthias Franz.

**Visualization:** Jan-Frederik Westermann.

**Writing – original draft:** Jan-Frederik Westermann.

**Writing – review & editing:** Tobias Müller, Matthias Franz.

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
