## [Decision Letter · Decision Letter 0]

31 Jan 2023

PONE-D-22-35210Measuring facial mimicry: Automated facial coding vs. EMGPLOS ONE

Dear Dr. Westermann,

Thank you for submitting your manuscript to PLOS ONE. After careful consideration, we feel that it has merit but does not fully meet PLOS ONE’s publication criteria as it currently stands. Therefore, we invite you to submit a revised version of the manuscript that addresses the points raised during the review process. Although both reviewers indicated the relevance and value of your work, they also mention substantial critical comments that have to be addressed in a revision. 

We look forward to receiving your revised manuscript.

Kind regards,

Peter A. Bos

Academic Editor

PLOS ONE

Journal Requirements:

a) Did participants provide their written or verbal informed consent to participate in this study?

4. We note that Figures 2-5 includes an image of a [patient / participant / in the study]. 

Reviewers' comments:

Reviewer's Responses to Questions

**Comments to the Author**

1. Is the manuscript technically sound, and do the data support the conclusions?

Reviewer #1: Partly

Reviewer #2: Partly

2. Has the statistical analysis been performed appropriately and rigorously? 

Reviewer #1: No

Reviewer #2: Yes

3. Have the authors made all data underlying the findings in their manuscript fully available?

Reviewer #1: No

Reviewer #2: Yes

4. Is the manuscript presented in an intelligible fashion and written in standard English?

Reviewer #1: No

Reviewer #2: Yes

5. Review Comments to the Author

Reviewer #1: In the manuscript titled “Measuring facial mimicry: Automated facial coding vs. EMG,” Westermann et al. showed participants dynamic facial expressions of various emotions and simultaneously videotaped the participants and recorded the EMG of Zygomaticus major (ZM) and Corrugator supercilii (CS). They correlated EMG activities of ZM and CS with Affdex-estimated values of corresponding action units (AU) over 25 time points of 200 ms intervals. They reported that Affdex could reliably detect spontaneous facial mimicry of smile expressions. However, several factors adversely affected the valid estimation of Affdex's performance detecting spontaneous facial mimicry of frowning expressions.

I agree with the authors that video-based automatic FACS coding tools are very convenient. Several studies have already used such tools for detecting subtle facial movements, such as spontaneous facial mimicry (e.g., Drimalla et al. and Mauersberger et al. [1,2] using OpenFace), without empirical evidence on the validity of automated tools in detecting spontaneous facial mimicry, compared to the golden standard of EMG. This could potentially lead to biased conclusions on the mimicking behavior, particularly when made against a specific population, such as individuals with an autistic diagnosis. An adverse scenario could be that inferences were made that a particular population “lacks” spontaneous facial mimicry only because the measuring methods used (e.g., automated FACS coding) were not sensitive enough. I appreciate the authors’ efforts in evaluating such approaches' fundamental issue of measurement validity.

I found issues in study design, statistics, and inference-making problematic. The suitability of publication will be evaluated after a major revision properly addressing the following points.

Major points:

1. Currently, there are several automated FACS coders, including at least two open-source options, OpenFace [1,2] and Py-Feat (https://py-feat.org/), able to integrate with more state-of-the-art deep learning (neural network) models for the same purpose. Under the current situation, it is unclear why the authors chose Affdex, which requires a license, and provides fewer AU estimates, with the only option of the SVM model. The rationale behind the toolbox choice for estimation requires clarification and elaboration.

2. Since the authors only estimated the performance of Affdex, it is to be determined whether their results could be generalized to other automated FACS coding toolboxes trained with different algorithms. Please revise the title to clarify that the present study focused only on Affdex.

3. Line 95: “EMG is relatively invasive due to the use of measuring electrodes on the face and, a bias of the results is conceivable.” This sentence requires elaboration and supporting empirical evidence. Facial EMG using surface electrodes (just like EEG) did not necessarily irritate 4-month-old infants [3], and the so-called “bias of the results” was vague and unspecified.

4. Line 216: Please specify “the clinically defined thresholds” for subject exclusion to make this study reproducible. Also, in Table 1, please include the range of trait measurements.

5. In FACS, AUs were clearly numbered and named. The AUs that corresponded to ZM and CS were AU12, “Lip corner puller,” and AU4, “Brow lowerer,” which were also listed so in the publication of Affdex [4]. In the present manuscript, the authors used the term “smile action unit,” which was confusing and possibly misleading. The smiling action involves at least AU12 and AU6 (Cheek raiser) [5]. Did the authors use the average of multiple AU measures (AU12 and 6) to correlate with ZM activity or only AU12? Please clarify and follow the proper FACS terms.

6. Statistics:

a. For each emotion, participants were tested eight times (two age groups, two genders, and two runs). In the EMG data, separate left and right face measurements were averaged during the integration. In the Affdex data, data from stimuli of two genders were averaged. How did the authors deal with repeated measures of two age groups, runs, and genders in the EMG and Affdex data? Furthermore, the authors could use algorithms incorporating mixed effects to take advantage of the statistical power of repeated measures.

b. The authors performed 25 spearman correlations along the time series for each ZM and CS data. First, this approach treated time points as independent and identically distributed (IID), which might be used to infer that “when the measurement of CS is high, the measurement of AU4 by Affdex would be high”. This approach ignored the fact that both measurements were time series with characteristics of autocorrelation, trends, seasonality, etc. Second, the issue of alpha correction for multiple testing was not addressed. I suggest the use of algorithms testing whether “two time-series were correlated/showing the same trend,” which also considers the repeated measure design, such as the “Repeated measures correlation” (rmcorr R package), testing whether ZM correlated with AU12, and CS correlated with AU4, using “subject” as a random factor, include all repeated measures without averaging, and use the option “bootstrapping” for confidence interval estimation.

c. Since there was a noticeable time lag between the muscle activity (EMG) and the facial landmark changes (AU) with reasonable explanations, the authors should consider testing cross-correlations between EMG and AU time series. The authors should consider other statistical algorithms appropriate for testing their hypothesis.

d. Reports of statistical results should include test statistics (r), degrees of freedom, effect sizes, confidence intervals, and exact corrected p-values, as per PLOS One statistical guideline (https://journals.plos.org/plosone/s/submission-guidelines.#loc-statistical-reporting).

7. It is unclear how Affdex was trained for emotion expression classification. McDuff et al. [4] only described that “The emotion expressions are based on combinations of facial actions. This coding was built on the emotional facial action coding system”. Was the SVM algorithm also used for emotion expression classification based on the AU patterns?

8. Sato and Yoshikawa [6] reported latency of maximally 900 ms post-stimulus onset when using automated FACS, while the present study reported a 1200 ms latency for smile mimicry. Could this indicate a lower sensitivity of Affdex compared to other automated FACS?

9. There were several reasons why the angry mimicry was not detected by Affdex, one of which concerns that the placement of CS electrodes adversely affected the Affdex estimation. It is still being determined whether retouching the video data for electrode placements is valid data processing. In this case, the null results did not exemplify that Affdex “cannot detect” angry mimicry. Instead, in the study setup, the placement of electrodes confounded Affdex’s ability to estimate AU4. I suggest modifying the inference and claiming that the present study is not suitable for estimating Affdex’s performance in AU4, as discussed in the second half of the Discussion. I also agree with the authors that the initial validity evaluation of automated FACS should start with comparing the performance of human FACS coders and automated FACS coders.

10. While the data reporting included the data tables, the code for statistical analysis needed to be included; please also include the code for the study reproducibility.

Minor points:

11. After formatting the citation style, the authors did not check the manuscript's readability. In numerous instances, such as in line 72, “18 (18) showed that the presentation of happy faces led to an increase in EMG activity…” the sentence does not make sense. The first “18” should have been the author of reference No. 18, Dimberg, while “(18)” referred to reference 18. I suggest the authors check the manuscript thoroughly before resubmission.

12. Section 3.1: abbreviations of interviews and questionnaires were mentioned before the full terms were introduced. When introducing the full terms, such as the “Structured Clinical Interview for DSM-IV” in line 160, the abbreviations did not follow the full term. This also negatively affected readability.

13. Lines 492 & 499: The same sentence was repeated twice – “35 (35) studied a healthy cohort who imitated faces with maximum affect expression. Here, Affdex measured a maximum mean of….” I suggest the authors reinspect and reorganize this paragraph. See also point 11.

Bibliography

1. Baltrušaitis T, Mahmoud M, Robinson P. Cross-dataset learning and person-specific normalisation for automatic Action Unit detection. 2015 11th IEEE International Conference and Workshops on Automatic Face and Gesture Recognition (FG). 2015. pp. 1–6. doi:10.1109/FG.2015.7284869

2. Baltrušaitis T, Zadeh A, Lim YC, Morency L-P. OpenFace 2.0: Facial Behavior Analysis Toolkit. 2018 13th IEEE International Conference on Automatic Face & Gesture Recognition (FG 2018). 2018. pp. 59–66. doi:10.1109/FG.2018.00019

3. de Klerk CCJM, Hamilton AF de C, Southgate V. Eye contact modulates facial mimicry in 4-month-old infants: An EMG and fNIRS study. Cortex. 2018;106: 93–103. doi:10.1016/j.cortex.2018.05.002

4. McDuff D, Mahmoud A, Mavadati M, Amr M, Turcot J, Kaliouby R el. AFFDEX SDK: A Cross-Platform Real-Time Multi-Face Expression Recognition Toolkit. Proceedings of the 2016 CHI Conference Extended Abstracts on Human Factors in Computing Systems. San Jose California USA: ACM; 2016. pp. 3723–3726. doi:10.1145/2851581.2890247

5. Cordaro DT, Sun R, Keltner D, Kamble S, Huddar N, McNeil G. Universals and cultural variations in 22 emotional expressions across five cultures. Emotion. 2018;18: 75–93. doi:10.1037/emo0000302

6. Sato W, Yoshikawa S. Spontaneous facial mimicry in response to dynamic facial expressions. Cognition. 2007;104: 1–18. doi:10.1016/j.cognition.2006.05.001

Reviewer #2: The manuscript titled "Measuring facial mimicry: Automated facial coding vs. EMG" addresses an issue commonly encountered when attempting to apply automatic facial action coding to videos of faces only showing subtle expressions. The results are what is to be expected and the authors could highlight better in which situations automatic coding might be suitable and in which situations facial EMG is the method of choice. I'm presenting comments in the following in the order they emerged while reading through the manuscript.

The authors write that facial mimicry is part of emotion recognition. However, it is more that this is subject to understanding of applied methods and the interpretation thereof. Manipulating facial muscles itself produces facial muscle activity that can affect emotion processing rather than the 'prevention' of mimicry.

For a discussion on incongruence between visual and motor information due to facial muscle manipulations, see https://link.springer.com/chapter/10.1007/978-3-031-08651-9_17

https://www.ncbi.nlm.nih.gov/pmc/articles/PMC5997820/

Related to this first comment: While the introduction presents a good summary of the current state of research on facial mimicry, the wording should get adjusted a little to more accurately represent what the current evidence suggests rather than taking all evidence as definite truth and facts.

Maybe the authors also would like to acknowledge that facial mimicry is evidenced beyond happy and angry (positive and negative expressions): https://www.ncbi.nlm.nih.gov/pmc/articles/PMC7069962/

line 90 "Other mental disorders with relevant changes in facial mimicry" - this statement is vague, what are 'relevant changes'?

line 95 "EMG is relatively invasive" - actually, surface electrodes are considered non-invasive (as apposed to invasive techniques such as using needle electrodes). see: https://link.springer.com/chapter/10.1007/978-3-031-08651-9_17

The content of the following comment was hinted on by the authors but could use more elaboration and could be louder in the manuscript. (In our lab, we also attempted to compare facial EMG to automatic coding and ran into the exact same problems). The electrodes in the face covered by face-colouring quite likely impacted the detection threshold of the coding software. This is because the colouring hides small wrinkles and general skin movement that would otherwise be visible. This could contribute to later detection times from the videos, and particularly for the corrugator area where the electrodes cover much of the emotional information and only very minimal information becoming visible given the nature of the phenomenon investigated; in covert facial mimicry we expect barely visible facial expressions. In line with that, the authors should comment on whether any facial expressions were visible in the videos.

With the multitude of correlations conducted on a data sequence, a correction for multiple testing needs to be applied.

6. PLOS authors have the option to publish the peer review history of their article (what does this mean?). If published, this will include your full peer review and any attached files.

Reviewer #1: No

Reviewer #2: No

---

## [Author Response · Author response to Decision Letter 0]

5 Jun 2023

Dear Editors,

We would like to thank you for your letter, the helpful comments and the opportunity to resubmit a revised copy of our manuscript. 

Your constructive feedback helped us to improve our paper and to focus on the most important results of our study. In the following, we will respond to each point raised by the reviewers and will refer to the changed text passage.

Reviewer #1.

1. Currently, there are several automated FACS coders, including at least two open-source options, OpenFace [1,2] and Py-Feat (https://py-feat.org/), able to integrate with more state-of-the-art deep learning (neural network) models for the same purpose. Under the current situation, it is unclear why the authors chose Affdex, which requires a license, and provides fewer AU estimates, with the only option of the SVM model. The rationale behind the toolbox choice for estimation requires clarification and elaboration.

We explained our choice in more detail and added the following to the manuscript: 

Affdex is one of the most widely used automated facial coding software. It promises ease of use and synchronization with other psychophysiological measures. It can be used to synchronously measure and evaluate various psychophysiological signals. Should a validation for the measurement of facial mimicry be successful, complex experimental paradigms could thus be performed in a relatively user-friendly manner. Lines 107-112.

2. Since the authors only estimated the performance of Affdex, it is to be determined whether their results could be generalized to other automated FACS coding toolboxes trained with different algorithms. Please revise the title to clarify that the present study focused only on Affdex.

We have changed the title to “Measuring facial mimicry: Affdex vs. EMG” Line 4

3. Line 95: “EMG is relatively invasive due to the use of measuring electrodes on the face and, a bias of the results is conceivable.” This sentence requires elaboration and supporting empirical evidence. Facial EMG using surface electrodes (just like EEG) did not necessarily irritate 4-month-old infants [3], and the so-called “bias of the results” was vague and unspecified.

We have changed the sentence into the following: In addition, before attaching the electrodes, the skin of the subjects must be cleaned with alcohol and rubbed with an abrasive electrode paste. It is conceivable that this procedure is unpleasant for some subjects.Lines 100-102 

4. Line 216: Please specify “the clinically defined thresholds” for subject exclusion to make this study reproducible. Also, in Table 1, please include the range of trait measurements. 

We have added the clinically defined thresholds in the text and in Table 1 Lines 232-234; 239.

5. In FACS, AUs were clearly numbered and named. The AUs that corresponded to ZM and CS were AU12, “Lip corner puller,” and AU4, “Brow lowerer,” which were also listed so in the publication of Affdex [4]. In the present manuscript, the authors used the term “smile action unit,” which was confusing and possibly misleading. The smiling action involves at least AU12 and AU6 (Cheek raiser) [5]. Did the authors use the average of multiple AU measures (AU12 and 6) to correlate with ZM activity or only AU12? Please clarify and follow the proper FACS terms. 

We have clarified that Affdex uses the term “Smile” instead of “Lip corner puller” and the term “Brow furrow” instead of the term “Brow lowerer”. Regarding your feedback, we are now following the FACS terms and only using the terms “Lip corner puller” and “Brow lowerer”. Lines: throughout the text.

6. Statistics:

a. For each emotion, participants were tested eight times (two age groups, two genders, and two runs). In the EMG data, separate left and right face measurements were averaged during the integration. In the Affdex data, data from stimuli of two genders were averaged. How did the authors deal with repeated measures of two age groups, runs, and genders in the EMG and Affdex data? Furthermore, the authors could use algorithms incorporating mixed effects to take advantage of the statistical power of repeated measures.

We have now explained our procedure more clearly in line 157-162 and in line 371-376: 

b. The authors performed 25 spearman correlations along the time series for each ZM and CS data. First, this approach treated time points as independent and identically distributed (IID), which might be used to infer that “when the measurement of CS is high, the measurement of AU4 by Affdex would be high”. This approach ignored the fact that both measurements were time series with characteristics of autocorrelation, trends, seasonality, etc. Second, the issue of alpha correction for multiple testing was not addressed. I suggest the use of algorithms testing whether “two time-series were correlated/showing the same trend,” which also considers the repeated measure design, such as the “Repeated measures correlation” (rmcorr R package), testing whether ZM correlated with AU12, and CS correlated with AU4, using “subject” as a random factor, include all repeated measures without averaging, and use the option “bootstrapping” for confidence interval estimation.

We chose this approach because it allows us to see over time at what point the EMG measurement and the Affdex measurement correlate significantly with each other. It is true that this method is not optimally suited to correlate repetitive measurements with each other. First, we performed an alpha correction (Benjamini-Hochberg) as suggested. Second, we calculated repeated measures correlations (rmcorr) for each measurement section. We integrated the results into the manuscript. We have mapped all four plots here in the rebuttal letter. We decided to show only the first plot as an example in the manuscript and to add the other results in a table in results section. 

c. Since there was a noticeable time lag between the muscle activity (EMG) and the facial landmark changes (AU) with reasonable explanations, the authors should consider testing cross-correlations between EMG and AU time series. The authors should consider other statistical algorithms appropriate for testing their hypothesis.

We have tested cross-correlations as proposed with the following results: 

We consider the calculation of cross correlations as well as the repeated measures correlation as a useful method to support our hypothesis. The results of the cross correlations shown below demonstrate this. For the sake of clarity, we have decided not to include the cross correlation in the manuscript additional. We believe that the repeated measures correlations already contain sufficient explanatory power. If you find it useful to include the Cross Correlations as additional information in the manuscript, we would do so. 

d. Reports of statistical results should include test statistics (r), degrees of freedom, effect sizes, confidence intervals, and exact corrected p-values, as per PLOS One statistical guideline (https://journals.plos.org/plosone/s/submission-guidelines.#loc-statistical-reporting).

We have added the missing information in the results section of the manuscript.

7. It is unclear how Affdex was trained for emotion expression classification. McDuff et al. [4] only described that “The emotion expressions are based on combinations of facial actions. This coding was built on the emotional facial action coding system”. Was the SVM algorithm also used for emotion expression classification based on the AU patterns? 

We also only have the information from Duff et.al. Therefore, we also assume this statement.

8. Sato and Yoshikawa [6] reported latency of maximally 900 ms post-stimulus onset when using automated FACS, while the present study reported a 1200 ms latency for smile mimicry. Could this indicate a lower sensitivity of Affdex compared to other automated FACS?

In the work of Sato et.al. FACS coding was performed by human scorers trained in FACS coding. This work is very important to our work because it shows that human raters are able to detect lip corner puller activity more quickly than Affdex and to detect changes in the brow lowerer as responses to dynamic affect stimuli. Therefore we have included this aspect in discussion Lines 561-572.

9. There were several reasons why the angry mimicry was not detected by Affdex, one of which concerns that the placement of CS electrodes adversely affected the Affdex estimation. It is still being determined whether retouching the video data for electrode placements is valid data processing. In this case, the null results did not exemplify that Affdex “cannot detect” angry mimicry. Instead, in the study setup, the placement of electrodes confounded Affdex’s ability to estimate AU4. I suggest modifying the inference and claiming that the present study is not suitable for estimating Affdex’s performance in AU4, as discussed in the second half of the Discussion. I also agree with the authors that the initial validity evaluation of automated FACS should start with comparing the performance of human FACS coders and automated FACS coders.

We agree with you and have adjusted the discussion and conclusion accordingly.

10. While the data reporting included the data tables, the code for statistical analysis needed to be included; please also include the code for the study reproducibility.

We have now included the code.

Minor points:

11. After formatting the citation style, the authors did not check the manuscript's readability. In numerous instances, such as in line 72, “18 (18) showed that the presentation of happy faces led to an increase in EMG activity…” the sentence does not make sense. The first “18” should have been the author of reference No. 18, Dimberg, while “(18)” referred to reference 18. I suggest the authors check the manuscript thoroughly before resubmission.

We have corrected this.

12. Section 3.1: abbreviations of interviews and questionnaires were mentioned before the full terms were introduced. When introducing the full terms, such as the “Structured Clinical Interview for DSM-IV” in line 160, the abbreviations did not follow the full term. This also negatively affected readability.

We have corrected this.

13. Lines 492 & 499: The same sentence was repeated twice – “35 (35) studied a healthy cohort who imitated faces with maximum affect expression. Here, Affdex measured a maximum mean of….” I suggest the authors reinspect and reorganize this paragraph. See also point 11.

We have corrected this.

Reviewer #2

The authors write that facial mimicry is part of emotion recognition. However, it is more that this is subject to understanding of applied methods and the interpretation thereof. Manipulating facial muscles itself produces facial muscle activity that can affect emotion processing rather than the 'prevention' of mimicry.

For a discussion on incongruence between visual and motor information due to facial muscle manipulations, see https://link.springer.com/chapter/10.1007/978-3-031-08651-9_17

https://www.ncbi.nlm.nih.gov/pmc/articles/PMC5997820/

1.Related to this first comment: While the introduction presents a good summary of the current state of research on facial mimicry, the wording should get adjusted a little to more accurately represent what the current evidence suggests rather than taking all evidence as definite truth and facts.

Maybe the authors also would like to acknowledge that facial mimicry is evidenced beyond happy and angry (positive and negative expressions): https://www.ncbi.nlm.nih.gov/pmc/articles/PMC7069962/

We have adapted the introduction accordingly Lines 52-55; 76-89

2.line 90 "Other mental disorders with relevant changes in facial mimicry" - this statement is vague, what are 'relevant changes'?

We have adjusted this sentence. Line 95.

3.line 95 "EMG is relatively invasive" - actually, surface electrodes are considered non-invasive (as apposed to invasive techniques such as using needle electrodes). see: https://link.springer.com/chapter/10.1007/978-3-031-08651-9_17

We have changed the sentence into the following: In addition, before attaching the electrodes, the skin of the subjects must be cleaned with alcohol and rubbed with an abrasive electrode paste. It is conceivable that this procedure is unpleasant for some subjects.

4.The content of the following comment was hinted on by the authors but could use more elaboration and could be louder in the manuscript. (In our lab, we also attempted to compare facial EMG to automatic coding and ran into the exact same problems). The electrodes in the face covered by face-colouring quite likely impacted the detection threshold of the coding software. This is because the colouring hides small wrinkles and general skin movement that would otherwise be visible. This could contribute to later detection times from the videos, and particularly for the corrugator area where the electrodes cover much of the emotional information and only very minimal information becoming visible given the nature of the phenomenon investigated; in covert facial mimicry we expect barely visible facial expressions. In line with that, the authors should comment on whether any facial expressions were visible in the videos.

We agree with you and have adjusted the discussion and conclusion accordingly

5.With the multitude of correlations conducted on a data sequence, a correction for multiple testing needs to be applied.

We made Benjamini-Hochberg corrections for P values ≤ 0.05 and adjusted the figures accordingly. We have summarized the exact results in the results section.

We are very glad about your helpful and high quality comments. We are sure that your contributions have greatly enriched our work and we hope that you agree with our implementation. We are already looking forward to your next feedback!

Please address all correspondence concerning this manuscript to westermannjan@yahoo.de.

Thank you for taking this manuscript into consideration. 

Cordially 

Prof. Dr. M. Franz

---

## [Decision Letter · Decision Letter 1]

6 Jul 2023

PONE-D-22-35210R1Measuring facial mimicry: Affdex vs. EMGPLOS ONE

Dear Dr. Westermann,

Thank you for submitting your manuscript to PLOS ONE. After careful consideration, we feel that it has merit but does not fully meet PLOS ONE’s publication criteria as it currently stands. Therefore, we invite you to submit a revised version of the manuscript that addresses the points raised during the review process.

Although the reviewers were generally positive on the changes made, they still would like to see a few issues addressed. 

We look forward to receiving your revised manuscript.

Kind regards,

Peter A. Bos

Academic Editor

PLOS ONE

Journal Requirements:

Reviewers' comments:

Reviewer's Responses to Questions

**Comments to the Author**

1. If the authors have adequately addressed your comments raised in a previous round of review and you feel that this manuscript is now acceptable for publication, you may indicate that here to bypass the “Comments to the Author” section, enter your conflict of interest statement in the “Confidential to Editor” section, and submit your "Accept" recommendation.

Reviewer #1: (No Response)

Reviewer #2: (No Response)

2. Is the manuscript technically sound, and do the data support the conclusions?

Reviewer #1: Yes

Reviewer #2: Yes

3. Has the statistical analysis been performed appropriately and rigorously? 

Reviewer #1: Yes

Reviewer #2: (No Response)

4. Have the authors made all data underlying the findings in their manuscript fully available?

Reviewer #1: Yes

Reviewer #2: Yes

5. Is the manuscript presented in an intelligible fashion and written in standard English?

Reviewer #1: Yes

Reviewer #2: Yes

6. Review Comments to the Author

Reviewer #1: The authors have addressed the reviewers’ comments adequately.

There are three points left:

1. Fig. 1 to 4: The asterisk * highlighting significant Spearman correlations should be based on corrected p-values. Currently, the figures still highlight uncorrected p-values, so when there is no significant correlation after alpha correction for multiple comparisons in Table 5, there are still two asterisks in Fig. 4. The discrepancy between corrected p-values and asterisks also applies to Fig. 1 and 3.

2. Line 569-570: the problematic description “In addition, EMG is relatively invasive due to the use of measuring electrodes on the face, and bias in the results is conceivable” remained. Please revise it to make it consistent with the revised information in the Introduction.

3. After seeing the result of the repeated measures correlations and cross-correlations, I found that the cross-correlations provided statistical evidence about a lagged matching in temporal patterns between EMG and Affdex (augmenting the serial Spearman correlations in Fig. 1 to 4, from which such an inference could only be made with visual inspection and comparing the peaking time of two time-series). In contrast, repeated measures correlations collapsing all time points eventually said little beyond the relatively unspecific fact that the time series are correlated somehow. Ultimately, I leave it to the authors to decide whether they will include the repeated measures correlations, cross-correlations, or not at all.

Reviewer #2: I would like to thank the authors for revising their manuscript according to the reviewers' comments. However, some of my previous comments have not been fully addressed and I am highlighting here the necessary revisions:

The paragraph starting line 50 does not accurately present the current state of knowledge on the relationship between facial mimicry and facial emotion recognition. The results are contentious and should be presented as such. I urge the authors to tone down claims and statements that facial mimicry is involved in facial emotion recognition and present the evidence for a link as well as counter-evidence (e.g., https://pubmed.ncbi.nlm.nih.gov/11165351/).

I further caution the authors in using the term 'blocked' when describing facial muscle manipulations. The authors of the cited studies induced facial muscle activity that interfered with mimicry. The terminology of 'blocking' would be more appropriate for Botox studies.

In addition, I am listing further comments from reading through the revised version of the manuscript:

It is unclear what is meant by this sentence: "Similarly, an increase in mimic muscle activity due to, e.g., a task, may lead to lower accuracy in recognizing facial expressions". Please rephrase.

The study cited in line 96 [32] is incorrectly presented, as the results did not show group differences in the facial mimicry response.

line 102: Using abrasive gel is no necessity. I have never used abrasives on participant's faces, as the cleaning with an alcohol swap is sufficient for good data quality. This procedure is also more pleasant for participants.

In terms of the analysis approach, I suggest the authors follow one of the methods presented here: https://towardsdatascience.com/four-ways-to-quantify-synchrony-between-time-series-data-b99136c4a9c9

Given that the data presented in the manuscript constitutes a time series, it would be appropriate and more parsimonious to conduct time series analyses only.

line 550: I suggest to delete the part 'relatively invasive'. In addition, participants generally report getting used to the electrodes within minutes and not even 'feeling' them anymore over the course of the experiment.

7. PLOS authors have the option to publish the peer review history of their article (what does this mean?). If published, this will include your full peer review and any attached files.

Reviewer #1: No

Reviewer #2: No

---

## [Author Response · Author response to Decision Letter 1]

28 Jul 2023

Dear PLOS ONE-Team,

thank you for the helpful feedback and your support. For the minor review, we have uploaded the manuscript, the manuscript with track changes, and the rebuttal letter as requested. In addition, we have updated our figures and the cover letter.

As requested, we have reviewed our reference list and mentioned all changes in the rebuttal letter. we have not cited any retracted papers. 

Sincerly yours

Jan Westermann

---

## [Editor Report · Decision Letter 2]

10 Aug 2023

Measuring facial mimicry: Affdex vs. EMG

PONE-D-22-35210R2

Dear Dr. Westermann,

We’re pleased to inform you that your manuscript has been judged scientifically suitable for publication and will be formally accepted for publication once it meets all outstanding technical requirements.

Kind regards,

Peter A. Bos

Academic Editor

PLOS ONE
---

## [Editor Report · Acceptance letter]

15 Aug 2023

PONE-D-22-35210R2 

Measuring facial mimicry: Affdex vs. EMG 

Dear Dr. Westermann:

I'm pleased to inform you that your manuscript has been deemed suitable for publication in PLOS ONE. Congratulations! Your manuscript is now with our production department. 

Kind regards, 

on behalf of

Dr. Peter A. Bos 

Academic Editor

PLOS ONE